# Indonesian Mothers’ Experience of Their Daughter’s HPV Vaccination, and Factors Associated with Their Willingness to Recommend HPV Vaccination for Girls

**DOI:** 10.3390/vaccines12090998

**Published:** 2024-08-30

**Authors:** Setiyani Marta Dewi, Ariane Juliana Utomo, Linda Rae Bennett, Siswanto Agus Wilopo, Anna Barrett

**Affiliations:** 1Nossal Institute for Global Health, The University of Melbourne, Melbourne 3010, Australia; lbennett@unimelb.edu.au (L.R.B.); anna.barrett@unimelb.edu.au (A.B.); 2School of Geography, Earth and Atmospheric Sciences, The University of Melbourne, Melbourne 3010, Australia; ariane.utomo@unimelb.edu.au; 3Department of Biostatistics, Epidemiology and Population Health, Center for Reproductive Health, Faculty of Medicine, Public Health and Nursing, Universitas Gadjah Mada, Yogyakarta 55281, Indonesia; sawilopo@ugm.ac.id

**Keywords:** HPV vaccination, school-based vaccination, Indonesia, parental acceptance, parental knowledge, health education

## Abstract

In Indonesia, knowledge of parents’ experiences of their daughters’ HPV vaccination in school settings is limited. As Indonesia seeks to scale up its HPV vaccination program nationwide, parents’ perspectives hold important insights into how elements of the vaccination model can be sustained and improved. This study explored mothers’ experiences of their daughters’ HPV vaccination experiences, their knowledge of HPV risks and prevention, and factors associated with willingness to recommend HPV vaccination for girls. A cross-sectional online survey was conducted with 143 mothers of schoolgirls who had received HPV vaccination at schools in Yogyakarta and Jakarta. Multivariate logistic regression was used to assess factors associated with willingness to recommend HPV vaccination. Chi-square and independent *t*-tests were performed to assess relationships between variables. One-way ANOVA was used to test mean differences in knowledge scores among mothers with different education levels. Many respondents (62.4%) received key health information before their daughters’ HPV vaccination. Mothers’ average knowledge score was 6.07/10 (SD 2.35). Receiving satisfactory information was significantly associated with willingness to recommend HPV vaccination for girls to others. A significant association was found between mothers’ willingness to recommend HPV vaccination and having ever participated in cervical cancer screening themselves. Providing consistent health information that addresses the knowledge gaps and affirms the benefits and safety of HPV vaccines can improve the likelihood of mothers recommending HPV vaccination to others. The benefit of a synchronized approach to promoting primary and secondary prevention was supported by the findings.

## 1. Background

Recent data indicate Indonesia has one of the highest cervical cancer (CC) incidence rates in Asia at 24.4 per 100,000 women [1]. Furthermore, CC is the second most acquired cancer among Indonesian women, accounting for 17% of all new cancer cases among Indonesian females in 2020 [2]. This form of cancer has a very high mortality rate of around 21,000 out of 36,000 new cases diagnosed annually. Despite this alarming situation, efforts at primary prevention via Human papillomavirus (HPV) vaccination have been slow to progress. Since 2016, an HPV vaccination demonstration program offered free vaccination with two shots of the quadrivalent vaccine to 5th and 6th-grade schoolgirls in 5 out of 38 provinces. The coverage in the demonstration program is reported to be 90% to 99% of eligible girls between 2017 and 2018 [3]. However, in 2020, fewer than 1 in 10 eligible girls within the demonstration program were recorded as having received their second dose of the HPV vaccine, with the COVID-19 pandemic being the primary contributing factor for lowering the coverage between 2020 and 2022 when many children were schooled at home [4,5]. 

In 2022, Indonesia announced a nationwide HPV vaccination program for schoolgirls in fifth and sixth grades to be administered through the national routine childhood immunization program [6]. Subsequently, in November 2023, the country launched its national CC elimination plan for 2023–2030 with the target of vaccinating 90% of children aged 11–12, and women between the ages of 21 and 26 by 2030 [7]. Currently, there is no publicly available data on the progress of the scale-up program since its implementation only started at the beginning of 2024. The commitment to national expansion of free HPV vaccination for both boys and girls, as well as young women, signals a promising future for the elimination of CC in Indonesia. However, it remains unclear if future HPV vaccination uptake will reflect the high uptake in the demonstration program when scaled up to a nationwide program. Several quantitative and qualitative studies on parental acceptance of HPV vaccination for girls in Indonesia have reported high acceptance, despite a concurrent lack of awareness of HPV vaccination and knowledge of CC among parents [3,8,9,10,11]. While it is reassuring to see high acceptance of HPV vaccination among Indonesian parents, one study with a sample of 484 parents from 33 schools in Jakarta also identified that vaccine hesitancy, particularly among affluent parents, might be on the rise due to misinformation [12].

A few studies on health education interventions related to HPV vaccination in both high-income and middle-income countries have concluded that providing parents with health information about the safety and benefits of HPV vaccination made a difference in improving knowledge and acceptance of HPV vaccination [13,14,15]. One study conducted with 506 parents of primary schoolgirls from rural Java found that in-person structured information sessions, combined with interactive discussions, worked well to improve parents’ knowledge of HPV vaccination in rural Indonesia where internet access is intermittent [15]. Systematic reviews of HPV vaccination implementation in low-and-middle-income countries (LMICs) and the United States also noted that educational interventions involving authoritative sources such as community leaders or health professionals were influential on parental acceptance and awareness of the HPV vaccine [13,16,17]. Additionally, having proactive recommendations from health providers has also been observed to improve parents’ confidence in consenting to their children’s HPV vaccination [18,19,20,21]. Regardless of methods, providing consistent and tailored information that addresses parents’ concerns, and the specific contexts that might exacerbate vaccine hesitancy, can assist in tackling vaccine misinformation [22,23].

In some Asian cultures such as Indonesia, Malaysia, and Bangladesh, it has been established that mothers (or female guardians and caregivers) are key sources for girls to learn about sexual and reproductive health issues, although sociocultural barriers and lack of knowledge have been observed as barriers to effective communication in some contexts [24,25,26]. Previous studies examining mother–daughter dyads regarding health behaviours have suggested that the mother–daughter relationship is a potential social and educational asset for promoting positive health behaviours to both mothers and their daughters [27,28]. Meanwhile, the literature on mother–daughter-targeted CC interventions has indicated that school-based HPV vaccination can also be a successful entry point to improve CC screening acceptance among women [21,29]. Particularly, providing information/education regarding CC screening for mothers/female caregivers of schoolgirls participating in school-based HPV vaccination is associated with better knowledge of CC prevention, and improved screening uptake when pathways to screening are made available [30,31]. 

Currently, in Indonesia, there is limited knowledge regarding parents’ experiences of their daughters’ HPV vaccination in school-based settings, and parents’ perspectives hold important insights into how core elements of this vaccination model can be improved in Indonesia. This article provides insight into parental acceptance of HPV vaccination for girls post-vaccination, where most prior studies on parental acceptance have been pre-vaccination [10,11,12,32]. Understanding parental acceptability is especially important given Indonesia’s ambitious goal to expand HPV vaccination nationwide. Previous studies on vaccination intention in Indonesia and the United States have found that community attitudes predicted positive health behaviours toward vaccination [23,32,33]. Particularly, knowledge of other people receiving the HPV vaccine is strongly associated with HPV vaccination uptake [34]. This article addresses a gap in the literature by exploring mothers’ knowledge of HPV, HPV vaccination, and CC prevention; their experience of their daughters’ HPV vaccination at school; and how these factors are correlated with mothers’ willingness to recommend HPV vaccination to others. The data analysed here are drawn from an online survey conducted with respondents from schools that participated in the HPV vaccination demonstration program between 2019 and 2021. Additionally, this analysis is integral to a larger study that also explored Indonesian schoolgirls’ experiences of HPV vaccination given through the demonstration program, which is beyond the scope of this article and has been published elsewhere [35].

## 2. Methods

### 2.1. Study Design

Data were collected between December 2021 and March 2022 via a cross-sectional online survey from mothers and or female caregivers of schoolgirls in grades 5 and 6 (approximately ages 11–14) who had received at least 1 dose of HPV vaccine through the school-based HPV vaccination program. The use of online methods became more common during the COVID-19 pandemic, especially in Indonesia where large-scale social mobility restrictions prohibited mass gatherings in public spaces including schools [36]. Hence, for both ethical and health reasons, this study was conducted entirely online.

### 2.2. Participants and Recruitment

The research team worked with primary health centres in Jakarta and Yogyakarta to generate a list of schools participating in the school-based HPV vaccination (schoolgirls who obtained their HPV vaccination through the demonstration program generally received the first dose of HPV vaccine in 5th grade and the second dose in 6th grade.). Potential participants were then recruited through public and private primary schools who agreed to participate in this study. A total of 20 public and private schools in Jakarta and Yogyakarta were approached, and 15 chose to participate in the study. Potential respondents were then recruited through participating schools that agreed to distribute an electronic flyer (e-flyer) to a minimum of 25 eligible mothers of 5th and 6th graders through parent–teacher WhatsApp groups (the use of WhatsApp groups to exchange information and provide community updates is commonplace in both private and public settings in Indonesia). The total number of mothers who received the e-flyer cannot be confirmed because it was impossible to monitor the activity on WhatsApp groups. The e-flyer contained key information such as the study objectives, researcher contact details, respondent inclusion criteria, and a link to the electronic plain language statement form with further details about this study. 

## 3. Materials

The online survey consisted of five thematic sections including: 

### 3.1. Socio-Demographic Information

To understand the demographic profile of mothers who participated, we collected data on their age, highest level of education, economic status and city of residence. Age was measured on a continuous scale, while city of residence was categorised as 1 for Yogyakarta and 0 for Jakarta. The economic status variable was divided into three categories: income < IDR 2.6 million is categorized as low, IDR 2.6–6 million is middle, and >IDR 6 million is high. The education variable was categorised into 3 categories, with 1 representing completion of primary and/or junior high school, 2 representing completion of senior high school, and 3 representing completion of higher education. 

### 3.2. Daughters’ HPV Vaccination Experience

This section explored HPV vaccination coverage among respondents’ daughters, and their mothers’ experience related to the provision of health information, the consent procedure, HPV vaccination side effects, and willingness to recommend HPV vaccination to others, using a total of 6 questions. One question assessed schoolgirls’ HPV vaccination status with a question, “how many times did your daughter receive HPV vaccination at school?”. The responses were categorised as 1 time, 2 times, and don’t remember. A total of 3 questions were included to assess the provision of key health information from health workers or schools before daughters’ HPV vaccination. Respondents were asked to answer questions about (1) HPV infections and CC, (2) HPV vaccine and potential side effects, and (3) daughters’ experience of side effects. A question related to mothers’ consent before vaccination was also included, and responses were then categorised as yes, no, and don’t remember.

To assess the impact of daughters’ HPV vaccination experiences on mothers’ attitudes regarding HPV vaccination, respondents were asked the following question: “Based on your daughter’s HPV vaccination experience, would you recommend HPV vaccination for girls to your friends or relatives?”. Responses were then coded as a binary yes/no. The yes response was defined as a respondent’s willingness to recommend HPV vaccination based on her daughter’s HPV vaccination experience. A no response meant a respondent was unwilling to recommend or unsure about recommending HPV vaccination to others based on her daughter’s experience. The unwilling and unsure responses were combined into a single category due to the small number of responses in both categories.

### 3.3. Respondents’ History of CC Screening

Respondents’ participation in CC screening was assessed with a question, “Have you ever had a Pap smear or VIA” with responses categorised as a binary yes or no. 

### 3.4. Knowledge of HPV, HPV Vaccination, and CC

Respondents’ knowledge of HPV, HPV vaccination, and CC were measured using 10 questions presented in true or false format. An incorrect answer scored 0, and a correct answer scored 1 point. The minimum total score was 0 and the maximum score was 10. Please refer to Table A1 in Appendix A for the correct responses.

### 3.5. Mothers’ Satisfaction with Health Information

To measure mothers’ satisfaction with health information received prior to their daughters’ HPV vaccination, two independent variables were transformed into one variable—called mothers’ satisfaction. First, two variables assessing whether a mother received key health information before her daughter’s HPV vaccination and whether she was satisfied with the information given were transformed into 1 variable where responses were recoded into 0 (did not receive information); 1 (received information and unsatisfied); 2 (received information and unsure about recommendation); and 3 (received information and satisfied).

## 4. Data Analysis

Out of 157 expressions of interest received through the QualtricsXM USA platform, 151 mothers consented to participate in the survey and 6 did not. Initially, a total of 151 responses were recorded as complete. After data cleaning, 143 responses were finally included, as 8 responses were excluded as they failed to meet respondent criteria or were mock responses by teachers and primary health centre staff.

Descriptive statistics to describe respondent profiles were produced in frequencies and percentages. Chi-square tests were conducted to assess associations between the city of residence and the provision of key health information before the daughters’ HPV vaccinations, as well as between history of CC screening and willingness to recommend HPV vaccination. Additionally, one-way ANOVA and independent *t*-tests were performed to assess continuous variables.

Stepwise regression analysis was conducted to assess factors associated with willingness to recommend HPV vaccination for girls to friends and family. The rationale for the strategy was to show that the provision of satisfactory health information is key to encouraging peer-to-peer promotion of HPV vaccination for girls, regardless of the mother’s knowledge level. The variables included in the analysis were willingness to recommend HPV vaccination to others (categorical) as a dependent variable; age (continuous); city of residence; education; knowledge of HPV vaccination and CC (continuous); and mothers’ satisfaction regarding provision of key health information as independent variables. The model was tested for goodness of fit and multicollinearity. Data were cleaned and analysed using SPSS V.28.

## 5. Results

### 5.1. Socio-Demographic Characteristics

Overall, there were 143 responses included in the analysis. The mean age of respondents was 40.12 years (SD 5.92). Of 143 mothers, 93.7% were married at the time of this study, 78.3% indicated their occupation as homemakers, 10.5% as private-sector employees, 4.2% as self-employed, and 7.0% as other occupations (i.e., civil servants, health workers, or preferred not to say). Additionally, most respondents came from either low-income backgrounds (45.5%), or middle-income backgrounds (34.3%) (see Table 1).

### 5.2. Mothers’ Experience of Daughters’ HPV Vaccination at School

#### 5.2.1. Provision of Key Health Information and Consent Procedures before HPV Vaccination

Overall, 62.4% of respondents reported receiving key information (key health information in this study is defined as information about the benefits, safety, and side effects of HPV vaccination) regarding HPV, HPV vaccination, and CC, while 37.6% did not receive any information. Furthermore, around 45% reported receiving information regarding potential side effects of the HPV vaccine (see Table 2). Chi-square tests showed no significant association between the provision of key health information and the city of residence. Meanwhile, most respondents (70.7%) indicated that they consented to their daughters’ HPV vaccination, and no significant association was found between mothers’ consent and city of residence.

#### 5.2.2. Experience of Side Effects and Willingness to Recommend HPV Vaccination for Girls

Most respondents (85%) reported that their daughters did not experience any side effects, while 8.50% reported mild side effects, i.e., redness at the injection site, fever, muscle pain, headache, sleepiness, nausea, and diarrhoea. The most common side effects experienced were redness at the injection site and fever. Additionally, of those who reported mild side effects, none sought medical attention. Furthermore, the findings also showed that based on their daughter’s HPV vaccination, most respondents (77.8%) were willing to recommend HPV vaccination for girls to friends and family. However, around 22% said they were either unsure or unwilling (see Table 2) to do so for reasons that were not explored in this study. A chi-square test was performed to assess relationships between respondents’ history of CC screening and willingness to recommend HPV vaccination and the results showed a significant positive association (*p*-value 0.004). 

#### 5.2.3. Knowledge of HPV, HPV Vaccination, and CC

The mean knowledge score for mothers in this study was 6.07 out of 10 (SD 2.35). A total of 43 respondents (30.7%) scored between 8 and 10, 62 (44.3%) scored between 5 and 7, and 35 (25%) had scores between 0 and 4. Questions with a high number of incorrect answers included HPV being transmissible for both men and women, CC being asymptomatic at early stages, CC screening still required after HPV vaccination, and CC only developing in married women (see Table 3).

Additionally, a one-way ANOVA test was performed to compare differences in the mean knowledge score at different levels of mothers’ education. As expected, the results showed that mothers with higher education (post senior high school) had significantly higher mean knowledge scores compared to those with only primary or secondary education, and those with high school education (see Table 4). Independent *t*-tests were also conducted to compare means between mothers with CC screening history and those with no history and found no significant difference in the mean knowledge scores between these groups (see Table 5). 

#### 5.2.4. Factors Associated with Mothers’ Willingness to Recommend HPV Vaccination for Girls

Multivariate logistic regression analysis demonstrated that after holding everything constant, knowledge of HPV vaccination and CC prevention, and receiving satisfactory information from health workers before HPV vaccination, both strongly predict mothers’ willingness to recommend HPV vaccination to friends and family members (see Table 6). The odds of mothers recommending HPV vaccination was 1.661 times for every one-unit increase in the knowledge scale. Therefore, a higher knowledge score is associated with a higher likelihood of recommending HPV vaccination to friends and family. Holding age, city of residence, and education level constant, for mothers who received and were satisfied with the information given, the odds of recommending HPV vaccination are seven times higher than for those who did not receive any satisfactory information. These positive associations persisted when knowledge and satisfaction were included in the same model.

## 6. Discussion

According to the WHO goals for the Elimination of Cervical Cancer as a Public Health problem by 2030, which Indonesia has committed to in its own National Elimination Plan, 90% of girls should have been vaccinated against HPV by the age of 15 [37]. This research provides crucial insights into how Indonesia can more efficiently achieve this goal by working with parents, and mothers in particular, to strengthen the uptake of the HPV vaccination program across the nation. High acceptance of school-based HPV vaccination was confirmed in this study, with around 77% of mothers willing to recommend HPV vaccination for girls to other females. Conversely, around 22% of mothers in this study did not confirm their willingness to recommend vaccination. Societal-wide acceptance of HPV vaccination will be crucial to reach the desired 90% uptake rate, and this study found that willingness to recommend HPV vaccination was strongly correlated with the provision of satisfactory key health information and good knowledge of HPV vaccination and CC prevention, which highlights the health promotion function of the vaccination program as a critical asset in achieving high acceptance as the program is scaled up.

The fact that 37.5% of mothers reported not receiving information prior to their daughters’ vaccination highlights a crucial missed opportunity in the demonstration program for achieving greater societal support for vaccination, and for ensuring mothers have the right to fully informed consent regarding their daughters’ vaccination. The current sites where the HPV vaccination demonstration program has been operational represent higher educational levels than in Indonesia as a whole. Thus, the positive correlation between mothers’ higher level of education and higher acceptance of vaccination indicates that targeted education via health promotion will be of crucial importance within remote and rural communities where formal educational levels are lower and HPV vaccination has yet to be introduced. The fact that having received information prior to daughters’ vaccination was a stronger predictor of willingness to recommend vaccination to others than mothers’ level of formal education indicates that women with lower formal education can be well equipped to make informed decisions regarding HPV vaccination and recommending it to others based on the specific information provided within the program. Thus, as the national program expands to reach regional and remote areas where formal educational attainment is typically lower, the health promotion function of the HPV vaccination program represents a key asset that has the potential to act as a leveller of educational differences across the nation, which in turn is important for achieving equity in access to HPV vaccination.

Despite bias in this study’s design towards mothers from schools chosen for the demonstration program and with active community involvement in the childhood school health program, the findings still indicate that 4 in 10 respondents did not receive adequate key health information about HPV and CC. Numerous studies from around the world discussing the acceptability of school-based HPV vaccination suggest high public acceptance of this approach but also emphasize that acceptability is highly contingent on strong community awareness of HPV vaccination and knowledge of HPV risks and prevention [16,21,38]. This study is the first to measure unmet needs for HPV and CC-related knowledge among Indonesian mothers involved in the vaccination program and to identify specific areas of information where women’s understanding was particularly low. In particular, health promotion and education interventions need to include information that was found lacking in this study sample. This information should include the risk of HPV infection in both men and women; the need for CC screening after HPV vaccination; the fact that CC is asymptomatic at early stages; what the symptoms are at more advanced stages; and the fact that CC affects all women regardless of marital status. The comprehensiveness and clarity of information provided to communities is imperative not only to ensure sufficient knowledge acquisition among parents and girls, in order to support high uptake but also to improve HPV vaccine acceptance for boys in the future, as outlined in the National Elimination Plan. 

This study also found that 70% of mothers had never experienced CC screening despite being in the appropriate age group for screening as per WHO recommendations. This low screening rate revealed there is both a need and opportunity for applying a more holistic and synergized approach to CC prevention that integrates secondary prevention via CC screening for mothers alongside primary prevention via the school-based HPV vaccination program. Although studies related to the integration of vaccination and CC screening remain limited, evidence from studies conducted in Peru, South Africa, and East Asia reported that a synergized approach can be beneficial for raising awareness of CC screening, especially among mothers and female caregivers of schoolgirls [29,30,31]. Additionally, this approach has been observed to hold strong potential in terms of improving HPV vaccination uptake, as well as awareness of HPV-associated cancers and their prevention when complemented by health education interventions through in-person sessions or distribution of information and education materials [21,29]

### Study Limitations

This study relied on self-reported data and most of the sample was constituted of mothers from three public schools in Jakarta and three public schools in Yogyakarta. Hence, selection bias should be considered when interpreting the findings. Furthermore, extrapolating the findings to other settings should be conducted with caution as the findings are not representative of the whole HPV demonstration program. Additionally, the sites where the HPV demonstration program has been active have higher educational levels and median incomes than Indonesia as a nation overall. Lastly, the quantitative nature of this study means that the research cannot provide an in-depth understanding of the respondents’ motivations or context, and further research of a qualitative nature is advisable to this end.

## 7. Conclusions

As Indonesia scales up its school-based HPV vaccination program, paying close attention to drivers of HPV vaccine acceptability within the wider community will be critical. This study has established that higher education levels and sufficient knowledge of HPV vaccination and CC prevention among women were strongly correlated with a higher tendency to recommend the vaccination to others. Consequently, incorporating structured and consistent health education interventions for parents of children receiving HPV vaccination at school can help improve and maintain vaccination acceptability beyond the demonstration program. At the same time, community-wide health promotion should be conducted for both sexes and people of different ages to address key information gaps and emerging misinformation about HPV and CC. An improved educational component in the school-based program should promote the achievement of informed consent and should be tailored to specific communities to cater to the educational profile of parents in those communities. Additionally, integrating the promotion of CC screening into the school-based HPV vaccination program can serve as an opportunity to improve both CC screening and HPV vaccination uptake, which will keep Indonesia on track with its National CC Elimination Plan and the WHO’s Global CC Elimination Strategy. Finally, if Indonesia seeks to achieve 90% HPV vaccination coverage in girls by 2030, consistent and comprehensive community health education, particularly for parents in less economically developed regions, should be part of the scale-up program focus. 

## Figures and Tables

**Table 1 vaccines-12-00998-t001:** Number of respondents by city and sociodemographic characteristics (N = 143).

Demographics	City		Total	
	Jakarta	Yogyakarta	N	%
Age				
29–39	27	28	55	38.5
40–49	25	30	55	38.5
50–57	7	2	9	6.2
Missing	15	9	24	16.8
Highest education level				
Higher education	13	21	34	23.7
Senior high school	45	41	86	60.1
Junior high school	10	6	16	11.2
Primary school	5	1	6	4.2
Missing	1	0	1	0.8
Economic status				
Low income	15	50	65	45.5
Middle income	40	9	49	34.3
High income	10	5	15	10.5
Missing	9	5	14	9.7
History of CC screening				
Yes	21	18	39	27.3
No	53	51	104	72.7
Had a friend/family member with CC experience				
Yes	6	3	9	6.3
No	68	66	134	93.7
Daughter’s Vaccination Status				
Number of HPV vaccines received by daughter				
1 dose	31	32	63	44
2 doses	34	27	61	42.7
Don’t remember	9	10	19	13.3

**Table 2 vaccines-12-00998-t002:** Number of respondents by city and responses to questions regarding daughters’ HPV vaccination.

Before HPV Vaccination				
	City		Total	
	Jakarta	Yogyakarta	N	%
Access to information regarding HPV, HPV vaccine, and CC				
*Respondent received information about HPV, HPV vaccine, CC from school or health worker* ***(N =* 141*)***				
Yes	48	40	88	62.4
No	24	29	53	37.6
*(IF NO) Interest in receiving written information? **(N =** * **53*)***				
Yes	21	25	46	86.8
No	1	3	4	7.50
Don’t know	2	1	3	5.70
*Respondent received information regarding the HPV vaccine side effects **(N =** * **138*)***				
Yes	34	29	63	45.7
No	38	37	75	54.3
*(IF NO) Interest in receiving written information **(N =*** **73*)***				
Yes	33	30	63	86.3
No	3	4	7	9.60
Don’t know	2	1	3	4.10
**Consent before HPV vaccination**				
*Consented to daughter’s HPV vaccination either verbally or in writing **(N =** * **140*)***				
Yes	51	48	99	70.7
No	10	14	24	17.1
Don’t remember	11	6	17	12.2
**Experience After HPV Vaccination**				
**Daughter’s experience of the HPV vaccine side effects**				
*Experience of side effects after receiving HPV vaccination **(N =** * **141*)***				
Yes	4	8	12	8.50
No	63	57	120	85.2
Don’t remember	5	4	9	6.30
**Recommending HPV vaccination for girls**				
*Willingness to recommend HPV vaccines to other female friends/relatives **(N =** * **140*)***				
Yes	59	50	109	77.8
No	6	8	14	10
Don’t know	7	10	17	12.2

**Table 3 vaccines-12-00998-t003:** Responses to questions regarding HPV, HPV vaccination, and CC.

		Correct n (%)	Incorrect n (%)
**1.**	Human papillomavirus can cause cervical cancer.	115 (80.4)	28 (19.6)
**2.**	Only women can get HPV infections.	28 (19.7)	114 (80.3)
**3.**	HPV is a sexually transmitted infection (STI).	85 (60.3)	56 (39.7)
**4.**	Pap smear or visual inspection with acetic acid can detect pre-cancerous cells.	109 (76.2)	34 (23.8)
**5.**	Cervical cancer can be cured if detected early.	122 (85.3)	21 (14.7)
**6.**	Only married women can develop cervical cancer.	77 (53.8)	66 (46.2)
**7.**	Women with cervical cancer sometimes have no symptoms.	68 (47.6)	75 (52.4)
**8.**	All HPV vaccines are effective protection against cervical cancer.	101 (70.6)	42 (29.4)
**9.**	The best time to get HPV vaccination is before one begins sexual contact.	82 (57.7)	62 (42.3)
**10.**	Regular Pap smear is not needed after one has been fully vaccinated against HPV.	72 (50.7)	70 (49.3)

**Table 4 vaccines-12-00998-t004:** Differences in mean knowledge scores among mothers with different levels of education.

Highest Level of Education Attained	n	Mean Estimate	95% CI
Primary and secondary high school	22	5.50	4.36–6.64
Senior high school	82	5.67	5.18–6.16
Higher education	31	7.35	6.58–8.13

**Table 5 vaccines-12-00998-t005:** Differences in mean knowledge scores between women with and without a history of CC screening.

Category	n	(M ± SD)	*p*-Value
Had CC screening	37	(6.6 ± 2.0)	0.143
No CC screening	103	(5.8 ± 2.4)	

**Table 6 vaccines-12-00998-t006:** Logistic regression results for factors associated with mothers’ willingness to recommend HPV vaccination to others.

Variable	Model 1	Model 2	Model 3	Model 4
	**Exp (B)** **(95% CI)**	**Exp (B)** **(95% CI)**	**Exp (B)** **(95% CI)**	**Exp (B)** **(95% CI)**
**Mother’s age**	0.948 (0.874–1.030)	0.922 (0.839–1.043)	0.950 (0.866–1.040)	0.915 (0.822–1.020)
**City of residence—Jakarta** **(Ref: Yogyakarta)**	1.261 (0.496–3.209)	1.279 (0.448–3.655)	1.218 (0.449–3.308)	1.466 (0.455–4.724)
**Highest education–Primary or Secondary level** **(Ref: Higher education)**	1.024 (0.249–4.205)	0.651 (0.133–3.182)	1.027 (0.224–4.702)	0.448 (0.066–3.019)
**Highest education—High School level** **(Ref: Higher education)**	1.385 (0.267–7.173)	0.533 (0.076–3.741)	2.894 (0.467–17.917)	0.961 (0.098–9.456)
**Total knowledge scores**		**1.661 *** ** (1.293–2.134)		**1.748 *** ** (1.320–2.314)
**Satisfaction with the given information** **(Ref: did not receive any or satisfactory information)**			**7.308 *** ** (2.448–21.820)	**9.871 *** ** (2.688–36.242)

*p*-value: *** < 0.001. Model 1: adjusted for age, city, and highest education. Model 2: adjusted for age, city, highest education, and total knowledge scores. Model 3: adjusted for age, city, highest education, and satisfaction with given information. Model 4: adjusted for age, city highest education, total knowledge scores, and satisfaction with given information.

## Data Availability

The data presented in this study are not available for public access.

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
