# Peer review of "Indonesian Mothers’ Experience of Their Daughter’s HPV Vaccination, and Factors Associated with Their Willingness to Recommend HPV Vaccination for Girls"

_vaccines, 2024, doi:10.3390/vaccines12090998_

Round 1

Reviewer 1 Report

Comments and Suggestions for Authors

The paper by SM Dewi et al. discusses an interesting topic relative to the retrospective point of view of mothers of adolescent Indonesian girls who were vaccinated against human papillomavirus (HPV). This angle of view is uncommon and original. The findings of the authors may contribute to better implement the HPV vaccination programs in adolescents and help to increase the adherence of young people and their families to this measure of Public Health that still needs to be actively sustained.

The paper is easy to read, written in a good English language and well-documented.

A few points merit however to be clarified:

-      The survey was performed more than two years ago. The Covid-19 may have delayed the redaction of this paper. As HPV vaccination is a topic that evolves very quickly, it would be useful to explain the changes, if any, of the Indonesian program of HPV immunization in the meantime. It would also be interesting to explain how the main findings reported in this paper have been taken into consideration for improving this program.

-          What is the range of age of schoolgirls belonging to grades 5 and 6 of the Indonesian educational system?  

-          Paragraph 3.4: a reference to Annex 1 must be done for the 10 questions.

-          Results: are the demographics given in Table 1 representative of the Indonesian population in terms of employment or socio-economic background? This point is important for indicating whether the observed results can be applied to the Indonesian population (in totality or in part).

-          For Table 1 and Table 2, the differences are given between the Jakarta and the Yogyakarta populations. For people living outside this area, is it useful to show these two sets of results separately since no major differences were observed between the two groups? If there is no clear justification to separate the data, I suggest to group them in a single category. The two Tables could also be grouped into a single one with demographics, data regarding the mother history and data relative to daughter’s vaccination.

-          In Table 1, why are there so many missing data (26) regarding the age of mothers? It seems to be the easier parameter to collect!

-          Paragraph 5.2.2: it could be interesting to give more details about the mild side-effects that were observed. Another limiting point is the absence of analysis of the reasons that conducted 22% of the mothers not to recommend vaccination to friends’ daughters.

In conclusion, this paper represents a valuable contribution to the implementation of HPV vaccination.

Author Response

Comment 1: The survey was performed more than two years ago. The Covid-19 may have delayed the redaction of this paper. As HPV vaccination is a topic that evolves very quickly, it would be useful to explain the changes, if any, of the Indonesian program of HPV immunization in the meantime. It would also be interesting to explain how the main findings reported in this paper have been taken into consideration for improving this program.

Response: Thank you very much for your feedback, we have added a sentence in the introduction section (lines 53-54) explaining the absence of public data on the progress of the scale-up program as the scale-up program has only been implemented nationwide at the start of 2024.

Additionally, we have not yet had the opportunity to disseminate our findings to national and local stakeholders in Indonesia. However, it is a part of our dissemination plan to inform the relevant stakeholders in Indonesia of our findings. In particular, we will seek an opportunity to collaborate with primary health centres in Indonesia to develop information and education materials for parents and children that address the gaps in HPV risks and prevention knowledge.

Comment 2: What is the range of age of schoolgirls belonging to grades 5 and 6 of the Indonesian educational system? 

Response: In the Indonesian education system grades 5 and 6 range between ages 11 and 14. This information has been added in the method section par.1 (lines 118-119).

Comment 3: Paragraph 3.4: a reference to Annex 1 must be done for the 10 questions.

Response: This information has been added in paragraph 3.4 (line 176).

Comment 4: For Table 1 and Table 2, the differences are given between the Jakarta and the Yogyakarta populations. For people living outside this area, is it useful to show these two sets of results separately since no major differences were observed between the two groups? If there is no clear justification to separate the data, I suggest to group them in a single category. The two Tables could also be grouped into a single one with demographics, data regarding the mother history and data relative to daughter’s vaccination.

Response: A distinction has been made regarding the location of the samples because of the differences in the two cities. Jakarta is the country’s capital and is a global mega-city with a population size of around 11.5 million, where people are more exposed to global discourses about sexual and reproductive health, and is typically thought of as the most progressive city in Indonesia. Yogyakarta, on the other hand, is a much smaller city with a population of around 3.7 million people, it has a large population of tertiary students and is often considered a “university town”. However, culturally Yogyakarta is considered to be far more traditional in its adherence to Javanese cultural values. What the two locations have in common is that investment in health services is at the high end compared to other parts of the country, and education levels for both populations are also higher than in other locations. These positive characteristics underlie the choice of the two cities for the HPV demonstration program.

As for the sociodemographic table itself, we have simplified the table by collapsing all the subsample percentages into total percentages for each row, as per the suggestion from reviewer 3. 

Comment 5: In Table 1, why are there so many missing data (26) regarding the age of mothers? It seems to be the easier parameter to collect!

Response: Indeed, we thought it would be the most straightforward question to answer. Unfortunately, many respondents opted not to respond to this question. 

Comment 6: Paragraph 5.2.2: it could be interesting to give more details about the mild side-effects that were observed. Another limiting point is the absence of analysis of the reasons that conducted 22% of the mothers not to recommend vaccination to friends’ daughters.

Response: Thank you for your suggestion, we have now added a sentence in lines 232-234 about the most common type of side effects experienced and that no respondent who reported mild side effects sought medical attention. Unfortunately, we cannot elaborate further as we did not collect more information beyond what we have presented. 

Reviewer 2 Report

Comments and Suggestions for Authors

In order to improve the manuscript, here are some critics and comments that the authors should respond.

In Table 2 did the side effects are really significant because the results from respondents are splitting.

The results show that the age factor is more significant when referred to low-, medium- or high-income. Correlation with the education should be discuted in Asia and not restrictively in Indonesia!  

P,9, l.276: the authors are using *elimination* of cervical cancer (CC). It is abusive because the use of this word is political, here we are in science, the use of *control* is relevant. I might agree that the authors responded to Spagnoletti et al (2021) or relevantly improved the fight against HPV in Asia but here the manuscript is restrictively scientific. 

I prefer that the authors fully documented how safe are the HPV vaccines. The links with CC could be added in the title (rather factors….) because the authors have included it in the studies.

Overall the reviewers can learn more about Indonesians against vaccines. Notably, how safe they are against HPV. After all, the good of this manuscript is about the efficiency of HPV vaccines that are not harmfull.

Author Response

Comment 1: In Table 2 did the side effects are really significant because the results from respondents are splitting.

Response: Thank you for your question, the subsample percentages columns have been collapsed into total percentages as per suggestion from reviewer 3. The total percentage for experience of mild side effects is 8.5% out of 141 respondents.

Comment 2: The results show that the age factor is more significant when referred to low-, medium- or high-income. Correlation with the education should be discuted in Asia and not restrictively in Indonesia!  

Response: Unfortunately, we are unable to interpret the first comment meaningfully and thus unable to respond. Secondly, while the comment regarding the correlation about education is a little hard to understand, we believe it may refer to our discussion of findings where we affirm that targeted education via health promotion that reflects the specific needs of different communities will be essential to the ongoing success of the program in Indonesia. As we discuss the importance of education being tailored to the needs of different and diverse communities, we do not agree with the reviewer that our results can or should be generalised to "Asia." 

Comment 3: P,9, l.276: the authors are using *elimination* of cervical cancer (CC). It is abusive because the use of this word is political, here we are in science, the use of *control* is relevant. I might agree that the authors responded to Spagnoletti et al (2021) or relevantly improved the fight against HPV in Asia but here the manuscript is restrictively scientific. 

Response: We acknowledge the reviewer's discomfort with the term "elimination," however we see this as the personal opinion of the reviewer that does not align with international best practice, and thus request that we keep the term in the manuscript. Additionally, we have used the language of elimination not by choice but rather because it is the official language that WHO and the Indonesian government used in their policies and strategic plans. The word elimination is interpreted in a highly positive way among people working in the cervical cancer field - precisely because this is one of the very few cancers that can be prevented and thus elimination is a possibility that exists for cervical cancer and not most other cancers.

Comment 4: I prefer that the authors fully documented how safe are the HPV vaccines. The links with CC could be added in the title (rather factors….) because the authors have included it in the studies.

Response: With regard to the request that we document the generalised safety of the HPV vaccine, this is significantly beyond the scope of the study, and we cannot include data that we did not collect. Additionally, there is a huge number of clinical studies including some from Indonesia that establish the safety of the HPV vaccine.

Reviewer 3 Report

Comments and Suggestions for Authors

The topic is significant, but the generalizeability of the study is limited because of sample limitations.  These are noted in the limitations section and the authors note the sample was more educated and had a somewhat higher socioeconomic level than the rest of Indonesia.  The testing of variable relationships is of primary interest in this study as a result of the sample limitations.

The finding of the study are not surprising, but they are important.  Receiving desired information and actual knowledge about HPV were both associated with a higher willingness to recommend for HPV vaccination.  These two variables make a significant difference in the mothers/caregivers willingness to recommend HPC vaccination to significant others.  The test sample is a fairly favorable setting for acceptance of HPV vaccination as the families in the sample are more educated and are somewhat better off economically than the Indonesian sample as a whole.   As information satisfaction and actual knowledge had important impacts on HPV vaccine acceptance in this sample, they are likey to have an even more important impact on less informed and less economically developed regions.  If Indonesia is to meet its 2030 goals of eliminating the HPV virus with 90% vaccine acceptance, community information programs by the schools and health personnel will need to be enhanced and focused on.  This connection is made in the conclusion, but deserves additional emphasis at the end of the article.

Here are some changes I would recommend for a resubmitted paper.

First, the Table Presentation of data in the Tables needs some reworking because it is confusing at present.  There are few differences between the two subsamples on demographic variables, other than income.  The columns in Table 1, for instance, include percentages for the subsamples out of the total sample and then have a column total percentage of 100%, when the figures above do not add to 100%.  I recommend collapsing the subsamples and only include a chart footnote when the samples by city/region indicate a discrepancy. 

Here are two errors in the text of the article that should be corrected.
The authors note that the present study was part of a larger study that has been previously published-the list article 34 as the reference publication, but it is clear in my reading that it is actually reference 35.
-In line 283 of the manuscript, the authors write that 33% of the sample were not willing to recommend vaccination for girls.  The appropriate number should be 23%.

My final comment is a question to the researchers-Could the number of doses that daughters received-one or two doses be an additional dependent variable to analyze?  Was the fact that over 40% of the girls only received 1 dose a factor of scheduling, or something else?  As some 77% of the mothers/caretakers were willing to recommend the HPV vaccination to kin and friends, it may not reflect a gap in vaccination efforts, but I thought it was worth posing this as a question.

Author Response

Comment 1: The finding of the study are not surprising, but they are important.  Receiving desired information and actual knowledge about HPV were both associated with a higher willingness to recommend for HPV vaccination.  These two variables make a significant difference in the mothers/caregivers willingness to recommend HPC vaccination to significant others.  The test sample is a fairly favorable setting for acceptance of HPV vaccination as the families in the sample are more educated and are somewhat better off economically than the Indonesian sample as a whole.   As information satisfaction and actual knowledge had important impacts on HPV vaccine acceptance in this sample, they are likey to have an even more important impact on less informed and less economically developed regions.  If Indonesia is to meet its 2030 goals of eliminating the HPV virus with 90% vaccine acceptance, community information programs by the schools and health personnel will need to be enhanced and focused on.  This connection is made in the conclusion, but deserves additional emphasis at the end of the article.

Response: Thank you very much for your suggestion. We have added a closing sentence emphasising the need for consistent community health education as part of the scale-up program focus on the conclusion section (lines 370-373).

Comment 2: First, the Table Presentation of data in the Tables needs some reworking because it is confusing at present.  There are few differences between the two subsamples on demographic variables, other than income.  The columns in Table 1, for instance, include percentages for the subsamples out of the total sample and then have a column total percentage of 100%, when the figures above do not add to 100%.  I recommend collapsing the subsamples and only include a chart footnote when the samples by city/region indicate a discrepancy.

Response: Thank you for the suggestions, Table 1 and Table 2 have now been updated accordingly. Please see the updated tables in the revised manuscript.

Comment 3: The authors note that the present study was part of a larger study that has been previously published-the list article 34 as the reference publication, but it is clear in my reading that it is actually reference 35.

Response: Thank you for noticing the error, it is now has been corrected.

Comment 4: In line 283 of the manuscript, the authors write that 33% of the sample were not willing to recommend vaccination for girls.  The appropriate number should be 23%.

Response: The number has been corrected. 

Comment 5: My final comment is a question to the researchers-Could the number of doses that daughters received-one or two doses be an additional dependent variable to analyze?  Was the fact that over 40% of the girls only received 1 dose a factor of scheduling, or something else?  As some 77% of the mothers/caretakers were willing to recommend the HPV vaccination to kin and friends, it may not reflect a gap in vaccination efforts, but I thought it was worth posing this as a question.

Response: Many thanks for your questions. We conducted some preliminary analyses on several variables including the number of doses variable to check for associations with willingness to recommend HPV vaccination but found no association between the number of doses received and willingness to recommend HPV vaccination. Additionally, the study deliberately included both 5th graders  and 6th graders to ensure adequate take up of the survey as we wanted to understand the experiences of girls who received HPV vaccination for the first and second time. We have also added a footnote (par2.2, line 126) to provide more information on how the HPV vaccination program works in Indonesia at the time of the study (schoolgirls generally will receive their first dose in 5th grade and the second dose in 6th grade).